# Protective Effects of Medicinal Plant Decoctions on Macrophages in the Context of Atherosclerosis

**DOI:** 10.3390/nu13010280

**Published:** 2021-01-19

**Authors:** Eloïse Checkouri, Stéphane Ramin-Mangata, Nicolas Diotel, Wildriss Viranaicken, Claude Marodon, Franck Reignier, Christine Robert-Da Silva, Olivier Meilhac

**Affiliations:** 1Université de La Réunion, INSERM, UMR 1188 Diabète Athérothrombose Thérapies Réunion Océan Indien (DéTROI), 97490 Sainte-Clotilde, La Réunion, France; eloise.checkouri@gmail.com (E.C.); stephane.ramin-mangata@univ-reunion.fr (S.R.-M.); nicolas.diotel@univ-reunion.fr (N.D.); christine.robert@univ-reunion.fr (C.R.-D.S.); 2Habemus Papam, Food Industry, 97470 Saint-Benoit, La Réunion, France; franck.reignier@hotmail.com; 3Université de La Réunion, UMR Processus Infectieux en Milieu Insulaire Tropical (PIMIT) INSERM 1187, CNRS 9192, 97490 Sainte-Clotilde, La Réunion, France; wildriss.viranaicken@univ-reunion.fr; 4APLAMEDOM Réunion, 1, rue Emile Hugot, Batiment B, Parc Technologique de Saint Denis, 97490 Sainte Clotilde, La Réunion, France; claude.marodon@wanadoo.fr; 5CHU de La Réunion, CIC 1410, 97410 Saint-Pierre, La Réunion, France

**Keywords:** medicinal plant decoction, atherosclerosis, polyphenols, Reunion Island

## Abstract

Atherosclerosis is a hallmark of most cardiovascular diseases. The implication of macrophages in this pathology is widely documented, notably for their contribution to lipid accumulation within the arterial wall, associated with oxidative stress and inflammation processes. In order to prevent or limit the atherosclerosis damage, nutritional approaches and medicinal plant-based therapies need to be considered. In Reunion Island, medicinal plant-based beverages are traditionally used for their antioxidant, lipid-lowering and anti-inflammatory properties. The aim of our study was to assess the protective effects of eight medicinal plant decoctions in an in vitro model of RAW 264.7 murine macrophages exposed to pro-atherogenic conditions (oxidized low-density lipoproteins—ox-LDL—*E. coli* Lipopolysaccharides—LPS). The impact of polyphenol-rich medicinal plant decoctions on cell viability was evaluated by Neutral Red assay. Fluorescent ox-LDL uptake was assessed by flow cytometry and confocal microscopy. Activation of NF-κB was evaluated by quantification of secreted alkaline phosphatase in RAW-Blue™ macrophages. Our results show that medicinal plant decoctions limited the cytotoxicity induced by ox-LDL on macrophages. Flow cytometry analysis in macrophages demonstrated that medicinal plant decoctions from *S. cumini* and *P. mauritianum* decreased ox-LDL uptake and accumulation by more than 70%. In addition, medicinal plant decoctions also inhibited NF-κB pathway activation in the presence of pro-inflammatory concentrations of *E. coli* LPS. Our data suggest that medicinal plant decoctions exert protective effects on ox-LDL-induced cytotoxicity and limited macrophage lipid uptake. Moreover, herbal preparations displayed anti-inflammatory properties on macrophages that can be of interest for limiting the atherosclerotic process.

## 1. Introduction

Cardiovascular diseases are among the leading causes of death worldwide [1]. Atherosclerosis is considered to be one of the main pathophysiological causes leading to a cardiovascular event. This process, causing progressive obstruction and stiffening of the arteries is favored by several risk factors such as smoking, hypertension, diabetes and dyslipidemia [2].

The first steps of atherosclerosis involve endothelial shear stress and low-density lipoprotein (LDL) deposits in the subendothelial space [3]. LDL oxidation takes place within the intimal compartment and promotes monocyte attraction, leading to their differentiation into macrophages [4,5,6]. The excessive and unregulated uptake of oxidized low-density lipoproteins (ox-LDLs) by macrophages triggers their transformation into foam cells and induces cell death (contributing to the necrotic core formation) [7,8,9,10,11]. Mechanisms of oxidative stress triggered by ox-LDLs and inflammation (cytokine secretion, apoptosis and necrosis) participate in the aggravation of atherosclerosis [12,13]. Macrophages play a crucial role in the development and the potential resolution of atherosclerosis, by participating in all steps of atherogenesis, from the initial lipid accumulation to the plaque rupture that leads to clinical events [14,15]. Targeting macrophage metabolism and response to oxidative stress and inflammation in atherosclerosis could thus represent a therapeutic option in the management of this pathology [16,17].

Polyphenols are the most abundant antioxidants in human diet. More than 8000 polyphenols have been identified and several studies highlight their bioavailability and bioactivity [18,19,20]. Moreover, numerous studies underline the beneficial effects of dietary polyphenols (notably flavonoids) and medicinal plant components in atherosclerosis [21,22,23]. In addition, polyphenols are extensively described in the literature for their potential protective effects on cytotoxicity, foam cell formation and cholesterol efflux in the context of atherosclerosis [24,25].

Reunion Island is a hotspot of biodiversity with 22 plants listed in the French Pharmacopeia, widely used by the population for antioxidant, lipid-lowering and anti-inflammatory properties [26]. Our previous study on eight medicinal plants used in Reunion Island revealed the presence of bioactive polyphenols (such as quercetin, gallic acid, mangiferin) and antioxidant properties on in vitro models (human red blood cells and murine preadipocytes 3T3-L1) [27]. Ethnopharmacological studies support the interest of medicinal plant antioxidant and anti-inflammatory activities in the management of different pathologies including atherosclerosis [28,29,30,31].

In this study, eight medicinal plant decoctions used in Reunion Island were evaluated for their possible contribution to atherosclerosis prevention and treatment: *Aphloia theiformis*, *Ayapana triplinervis*, *Dodonaea viscosa*, *Hubertia ambavilla*, *Hypericum lanceolatum*, *Pelargonium x graveolens*, *Psiloxylon mauritianum* and *Syzygium cumini*. Based on our previous study [27] reporting the composition and antioxidant effects of infusions and decoction of these eight plants, we hypothesized that the variety of polyphenols contained in these extracts could confer them interesting properties to limit the processes underlying atherogenesis. Medicinal plants with a higher polyphenol content, especially those containing quercetin, isorhamnetin, chlorogenic acid or kaempferol, could have an interesting bioactivity in this context [32,33,34,35,36]. The impact of the medicinal plant decoctions on macrophage cytotoxicity induced by ox-LDL and inflammation-induced NF-κB signaling pathway as well as on ox-LDL uptake was also investigated.

## 2. Materials and Methods

### 2.1. Raw Material

All raw material was provided by HABEMUS PAPAM Industry proceeded with the following industrial process: trimming, microwave dehydration, grinding and packaging in tea bags containing 1 g of plant powder. Industrial lots and corresponding GPS coordinates are described in our previous study [27].

### 2.2. Preparation of the Decoctions

Decoctions were prepared by adding one tea bag containing 1 g of plant to 1 L of room temperature (RT) distilled water and kept boiling for 30 min. The tea bags were removed and the decoctions were cooled at RT before storage at −80 °C until analysis. Characterization of the polyphenol contents and antioxidant activity of the decoctions are reported in our previous study [27].

### 2.3. LDL Isolation

LDLs were extracted from a pool of plasma from 5 healthy subjects obtained from the “French Blood Agency”, according to an agreement signed between Inserm and “Etablissement Français du Sang”, which governs the supply of blood-derived products, in compliance with the French Ethical laws. Plasma was adjusted to a density of 1.063 g/mL by addition of solid KBr (Bio basic Inc., Markham, ON, Canada), with gentle stirring. Beckman Ultra-ClearTM sealing ultracentrifuge tubes (Beckman Coulter, Villepinte, France) were filled with an aqueous solution of KBr (d = 1.063) and the plasma (15 mL) was added underneath to obtain a discontinuous density gradient. Finally, 3 mL of distilled water were added before sealing. Samples underwent an ultracentrifugation at 252,000 *g* for 20 h, at 10 °C. Yellow LDL fractions were collected with a 10 mL syringe and followed a desalting step using centrifugal devices (50 mL centricon, 10 kDa cutoff, Millipore, Merk, Darmstadt, Germany) with 5 washed with PBS after successive centrifugations (4000× *g*/20 min). The purity of LDL was assessed by SDS-PAGE followed by Coomassie staining in order to detect potential protein contaminants. LDL protein concentration was quantified by using DC Protein Assay kit (Life Science, Bio-rad, Marnes-la-Coquette, France). LDL were stored at 4 °C in the dark and sterilized by filtration on a 0.2 µm membrane before use in cell culture.

### 2.4. LDL Oxidation

LDLs were diluted in 1X PBS to reach the concentration of 2 mg/mL and 5 µM final of CuSO_4_ (Thermo Fisher Scientific, Dardilly, France) was added before UV-C exposure for 2 h at RT, in order to obtain mildly oxidized LDL [37]. The degree of oxidation was assessed by quantifying thiobarbituric acid reactive substances (TBARS) according to Ohkawa et al. [38].The samples reached 18.07 ± 0.5 nmol/mg proteins [38]. Ox-LDL were stored at 4 °C in the dark and sterilized by filtration on a 0.2 µm membrane before use for cell culture.

### 2.5. Dil-ox-LDL Production

LDLs were first oxidized and then labelled with DilC_18_ (3,3′-dioctadecyloxacarbocyanine perchlorate) (Thermo Fisher Scientific, Dardilly, France), a lipophilic fluorescent probe that interacts with phospholipid layer membranes, leading to the formation of fluorescent Dil-LDL detectable at excitation and emission wavelengths of respectively 549 nm and 565 nm. Dil-ox-LDL production was done according to Xu et al., with slight modifications [39]. Ox-LDL were incubated overnight at 37 °C under gentle agitation with of Dil stock solution (3 mg/mL in DMSO) to reach a final concentration of 0.1 mg/mL. The excess of the unbound probe was filtered through a 0.2 µm membrane to discard undissolved crystals and then re-isolated by ultracentrifugation as described above (in KBr at 252,000 *g* for 20 h at 10 °C). The pink Dil-ox-LDL layer was recovered and desalted as described above. Filtered sterilized Dil-ox-LDL were added to cells at a final concentration of 100 µg/mL.

### 2.6. Cell Culture

RAW 264.7 (ATCC^®^ TIB-71™) murine macrophages were obtained from the American Type Culture Collection (ATCC©, Manassas, VA, USA). The culture medium included Dulbecco’s modified Eagle’s medium with 25 mM glucose, 10% heat-inactivated fetal bovine serum, 5 mM L-glutamine, 2 µg/mL streptomycin and 0.03 µg/mL penicillin (Pan Biotech, Dutscher, Brumath, France). Cells were maintained in a humidified 5% CO_2_ incubator at 37 °C for 4 days before the assay when a 70–80% confluency was reached.

### 2.7. Neutral Red Assay

Cell viability was assessed by measuring the intake of neutral red as described by Repetto et al., with slight modifications [40]. This test is based on the capacity of viable cell lysosomes to incorporate neutral red dye solution. Cells were first seeded at 250 × 10^3^ cells/mL in a 96-well plate for 24 h. Before incubation with LDL (native of oxidized), cells were serum-starved for 16 h. Stimulation with medicinal plant decoctions (40 µg/mL) and/or with native or oxidized LDL (50 µg/mL to 300 µg/mL) was carried out for 8, 16 and 24 h. The medium was removed and cells were washed twice with PBS. 100 µL of neutral red solution (40 µg/mL) was added and the cells were then incubated for 2 h at 37 °C. After 2 washes with PBS, neutral red destain solution (49.5% water, 49.5% ethanol, 1% glacial acetic acid) was added and the absorbance was read at 540 nm. Triton X100 (1%) was used as positive control for cytotoxicity (100% of cell death). Results are expressed as percentages of the control untreated cells.

### 2.8. Fluorescence and Confocal Microscopy

RAW 264.7 macrophages were seeded at 200 × 10^3^ cells/mL on coverslips in a 24-well plate and FBS-starved for 16 h before stimulation. Cells were washed with 500 µL of PBS and incubated for 4 h at 37 °C/5% CO_2_ with cell culture medium supplemented with Dil-ox-LDL. After the incubation, cells were rinsed 3 times with PBS and fixed with PBS containing 4% paraformaldehyde (PFA) for 15 min. Cells were rinsed twice with PBS, incubated with DAPI (4′,6-diamidino-2-phenylindol, 1 ng/mL, Life Technologies) for 10 min and rinsed twice with PBS. The coverslips were then recovered and sealed before analysis using a Nanozoomer scanner (Hamamatsu, Iwata City, Japan) and a Nikon C2Si confocal microscope (Nikon, Melville, IL, USA).

### 2.9. Flow Cytometry

RAW 264.7 cells were seeded at 250 × 10^3^ cells/mL in a 96-well plate. After 24 h, cells were serum-starved for 16 h. Cells were then stimulated with cell culture medium supplemented with Dil-ox-LDL for 4 h, lifted with Accutase™ (400 units/mL, Gibco, Thermo Fisher Scientific, Dardilly, France) for 15 min at 37 °C and transferred into a 96-well plate with U-bottom. Cells were fixed with 4% PFA for 15 min at RT and centrifuged at 700× *g* for 1 min. The supernatant was discarded and PBS was added. Cells were centrifuged at 700× *g* for 1 min and re-suspended in 100 µL PBS. Cellular uptake of Dil-ox-LDL was evaluated by measuring the mean phycoerythrin (PE-A) fluorescence on flow cytometer CYTOFLEX (Beckman Coulter, Villepinte, France). Fluorescence was expressed as percent of control.

### 2.10. Evaluation of NF-κB Activation

RAW-Blue™ cells (monocyte-macrophages) obtained from Invivogen cell culture collection (Invivogen, San Diego, CA, USA), were seeded at a density of 100 × 10^3^ cells/well in a 96-well plate. 20 µL of medium supplemented with decoctions (reaching 40 μg plant mash/mL of cell culture medium) was added to the cells. Cells were then immediately treated with *E. coli* lipopolysaccharides (LPS) Ultrapure (Invivogen, San Diego, CA, USA) at a final concentration of 0.5 µg/mL. After 24 h of incubation, 20 µL of medium was recovered and 80 μL of Quantiblue reagent (Invivogen, San Diego, CA, USA) was added. RAW-Blue™ cells stably express secreted embryonic alkaline phosphatase (SEAP) inducible by NF-κB. NF-κB activation was evaluated by the measure of the absorbance at 620 nm. Results were expressed as percentages of the control.

### 2.11. Statistics

Data were expressed as means ± SD of three independent experiments performed in triplicate. Statistical analysis was carried out using GraphPad Prism software (Version 6.01, © 1992–2012 GraphPad Prism Software, Inc., San Diego, CA, USA). Differences between the means were determined by the Bonferroni test and were considered as statistically significant for a *p* value < 0.05.

## 3. Results and Discussion

### 3.1. Medicinal Plant Decoctions Reduce ox-LDL Cytotoxic Conditions

Macrophages are involved in the different steps of atherogenesis from the initial accumulation of foam cells (forming fatty streaks) to more advanced atherosclerotic lesions in the necrotic core. Modulation of macrophage inflammation represents a potential target for the prevention of atherosclerotic plaque development and for limiting plaque rupture leading to clinical events [41,42].

In order to assess the influence of medicinal plant decoctions on macrophage viability, medicinal plant decoctions (40 µg plant mash/mL) were incubated with RAW 264.7 for 24 h (Figure 1a). Cells were first incubated with different concentrations of ox-LDLs (from 50 to 300 µg/mL) (Figure 1b). Once the cytotoxic concentration of ox-LDL was determined, the cytotoxicity was assessed at 8, 16 and 24 h (Figure 1c).

As presented in Figure 1a, medicinal plant decoctions alone did not significantly decrease cell viability (Figure 1a). In contrast, the significant increase of viable cells incubated with *H. ambavilla*, *A. triplinervis*, *H. lanceolatum*, *P. x graveolens* and *S cumini* could suggest an effect on cell proliferation. Figure 1b shows that incubation with 50 µg/mL ox-LDLs did not induce any cytotoxic effect; the cell viability was comparable to control non-stimulated cells. Using the same concentration of ox-LDLs (50 µg/mL), Ding et al. reported vascular smooth muscle cell proliferation as measured by the MTT assay [43]. This effect was not found in our experimental conditions, using macrophages. A significant decrease in cell viability was observed after the incubation with 200 and 300 µg/mL of ox-LDL (Figure 1b). The concentration of 200 µg/mL of ox-LDL leading to 50% of cell death was selected to assess the potential protective effect of medicinal plant decoctions.

Figure 2 shows the cell viability at 8, 16 and 24 h with cytotoxic concentration of ox-LDL (200 µg/mL) in the presence of medicinal plant decoctions.

Incubation with cytotoxic concentrations of ox-LDL for 8, 16 and 24 h resulted in decreased number of viable cells by 28.52 ± 4.40%, 35.47 ± 7.40% and 52.53 ± 2.43%, respectively. More than 80% of cell viability was observed in co-incubation with *A. theiformis*, *H. lanceolatum*, and *P. x graveolens,* maintaining 83.78 ± 10.91, 81.73 ± 2.60 and 80.98 ± 4.66% of viable cells at 24 h. The presence of polyphenols in the different decoctions may explain this bioactivity. As shown in Figure 2, most of the medicinal plant decoctions tested significantly decreased the cytotoxicity induced by ox-LDL at 8, 16 and 24 h. *H. ambavilla* and *A. triplinervis* displayed moderate effects that were only significant at 24 h and question the physiological relevance of this protection. An interesting aspect that should be taken into account is that *A. theiformis* which contains more than 98% of mangiferin and derivatives showed the higher protective effect. Several studies on mangiferin revealed its protective effects on cell survival both in vivo and *in vitro*, in models involving oxidative stress [44,45,46].

### 3.2. Medicinal Plant Decoctions Reduce the Dil-ox-LDL Uptake by RAW 264.7 Macrophages

Fatty streak formation is defined by the presence of lipid accumulation within the arterial subendothelial space (intima). In response to excessive accumulation of ox-LDL in the intima, macrophages take up modified LDL (including ox-LDL) in an uncontrolled fashion, leading to their transformation into foam cells. In contrast to native LDL endocytosis by the canonical LDL receptor, oxidized LDL uptake by scavenger receptors is not downregulated by a negative feedback loop [47]. Furthermore, lysosomal degradation of ox-LDL is partial and leads to intracellular lipid peroxide accumulation [48]. When foam cells are unable to accumulate more lipids, they trigger apoptosis and secondary necrosis processes. Several studies also showed the contribution of foam cells in inflammation, notably by secretion of cytokines and the release of cell debris [49,50]. Indeed, preventing foam cell formation is a therapeutic strategy to slow down the atherosclerosis process and some components such as quercetin and curcumin showed encouraging inhibitory effects [7,25,51].

As presented in Figure 3, quantitative analysis by flow cytometry (Figure 3a,b) showed that *H. lanceolatum*, *P. x graveolens*, *S. cumini* and *P. mauritianum* significantly decreased the Dil-ox-LDL uptake. Interestingly, *S. cumini* and *P. mauritianum* respectively inhibited Dil-ox-LDL uptake by 71.4 ± 8.67% and 74.3 ± 7.01%. Cells incubated with *A. triplinervis* showed similar profiles to that of control cells + Dil-ox-LDL, whereas *P. mauritianum* decoction, traditionally described for cholesterol lowering properties [26], was associated with a decreased abundance of intracellular Dil-ox-LDL (Figure 3c,d show immunocytofluorescence analyses of RAW 264.7 macrophages incubated with fluorescent Dil-ox-LDL with *A. triplinervis, P. mauritianum* decoctions or PBS). This property could support the traditionally described beneficial effect of *P. mauritianum* in cardiovascular diseases. However, this should be tested in vivo in order to assess its capacity to lower cholesterol levels (as suggested by ethnopharmacological data) and potentially to prevent foam cell formation. Our results suggest that medicinal plant decoctions could constitute a therapeutic option to prevent or reduce atherosclerosis progression, that would deserve in vivo experiments in pre-clinical models.

### 3.3. Medicinal Plant Decoctions Reduce Lipopolysaccharide-Induced NFkB Activation in RAW-Blue™ Macrophages

NF-κB signaling orchestrates different transcription pathways that regulate gene expression implicated in cytokine, chemokine and adhesion molecule production, cell survival and differentiation [52]. Atherosclerosis is associated with increased production of pro-inflammatory biomarkers such as C-reactive protein (CRP), Interleukin-6 (IL-6) or Interleukin-1β (IL-1β) [53]. Moreover, increased cytokine secretion by pro-inflammatory macrophages favors atherosclerosis progression. RAW-Blue™ macrophages were stimulated with different concentrations of ox-LDL in the presence of medicinal plant decoctions in order to assess their capacity to modulate NF-κB activation (monitored by the detection of secreted embryonic phosphatase alkaline -SEAP- in the medium, induced by NF-κB activation). In our experimental conditions, we were unable to show a significant increase in NF-κB activation after incubation with 100–200 µg/mL ox-LDL. Indeed, only ox-LDL stimulation at the cytotoxic concentration of 300 µg/mL (83% cell death) led to a significant increase of 61.1 ± 5.48% of SEAP release whereas 0.5 µg/mL LPS led to an increase of 165 ± 19.82% without altering cell viability. Because lipopolysaccharides (LPS) from bacteria such as *P. gingivalis* or *E. coli* are also known to exacerbate inflammation (IL-6, TNF-a secretion, NF-κB activation) and may be associated with atherosclerosis [30,54], we tested the macrophage response to LPS in the presence of medicinal plant decoctions. Our results presented in Figure 4, showed that *E. coli* LPS triggers SEAP levels to 209.1 ± 45.92% as compared to control. Medicinal plant decoctions (40 µg/mL) significantly reduced NF-κB activation induced by *E. coli* LPS (0.5 µg/mL).

The greatest inhibition of LPS NF-κB activation was observed for *P. x graveolens*, *H. lanceolatum* and *P. mauritianum* with, respectively, 93.0 ± 17.84, 86.4 ± 27.06 and 76.1 ± 8.33% of inhibition. Surprisingly, even if our results confirmed *H. lanceolatum* anti-inflammatory properties that were traditionally described, *P. x graveolens* showed stronger effects. Our results also underline the potential of most of our selected medicinal plant decoctions in the management of inflammation. Indeed, seven out of eight medicinal plant preparations significantly reduced SEAP release. A decrease of SEAP release was also observed for *A. theiformis* even if it was not statistically significant relative to the control (48.0 ± 24.99%; *p* < 0.08).

## 4. Conclusions

Medicinal plant decoctions traditionally used in Reunion Island could represent the basis of new therapeutic strategies aimed at preventing or limiting the progression of atherosclerosis [27,31]. Many medicinal plants from Reunion Island are rich in polyphenols, and their composition is described in our previous study [27]. Among these compounds, flavonoids such as quercetin could be, at least in part, responsible for the bioactivity of these plant-based herbal teas. Different studies also highlighted other polyphenols such as chlorogenic acid, kaempferol or mangiferin, reported to limit the cytotoxicity induced by fatty acids and oxidative stress [55,56,57,58,59,60]. This is consistent with the beneficial effects of decoctions from the eight selected plants from Reunion Island tested in this study, traditionally used for their antioxidant, anti-inflammatory and lipid-lowering properties. It cannot be excluded that these active ingredients act synergistically. The interest of the present study is to document the effects of complex mixtures of bioactive compounds (herbal teas rich in polyphenols) according to their traditional use in Reunion Island. Identification of the exact bioactive molecules in these plants is of interest and should be further investigated in complementary studies.

Our study has a number of limitations:-Murine macrophages were used and stimulated by human LDLs. This experimental design is commonly used in the literature. In addition, this cellular model was chosen as we plan to carry out in vivo studies to assess the bioactivity of decoctions in a mouse model of atherosclerosis. Furthermore, isolation of LDLs in mice is difficult to achieve due to limited blood volume [55].-The precise compounds and mechanisms responsible for the inhibition of the uptake of ox-LDL by *P mauritianum* and *S cumini* decoctions would deserve further investigation but cannot be explained by the presence of a single compound. Each decoction constitutes a complex set of molecules that can exert their effects synergistically. In addition, mechanistic aspects could to be investigated (for example, the study of the impact of decoctions on the expression of scavengers’ receptors).-In a physiological context, polyphenols can be transformed into secondary metabolites by intestinal of hepatic enzymes (including both host and microbiota enzymes).-In order to be able to evaluate the curative or preventive potential of the different decoctions tested, our in vitro study would benefit from being extended by studies on an in vivo model.-In summary, we demonstrated the effects of eight medicinal plant decoctions on ox-LDL cytotoxicity; we then showed that certain medicinal plant decoctions were able to significantly limit ox-LDL uptake and subsequent foam cell formation. Finally, we report the potential of decoctions to inhibit NF-κB activation, suggesting that herbal teas may prevent the activation of inflammatory signaling pathways.

## Figures and Tables

**Figure 1 nutrients-13-00280-f001:**
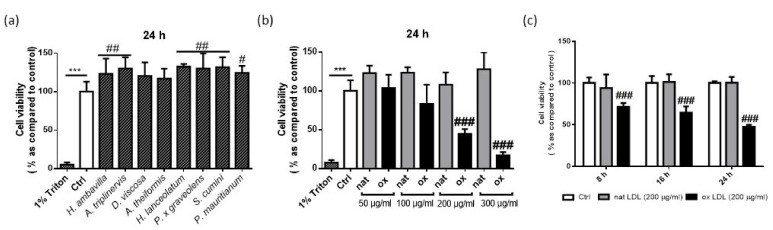
Cell viability of RAW 264.7 macrophages incubated with medicinal plant decoctions (40 μg plant mash/mL) (**a**), different concentrations of native (white) or oxidized low-density lipoproteins (LDL) (black) (**b**), cytotoxic concentration of ox-LDL (200 μg/mL) (**c**). Data shown are means ± SD of three independent experiments. *** *p* < 0.005 as compared to 1% Triton X-100; ^#^
*p* < 0.05; ^##^
*p* < 0.01, ^###^
*p* < 0.005 as compared to control (Ctrl).

**Figure 2 nutrients-13-00280-f002:**
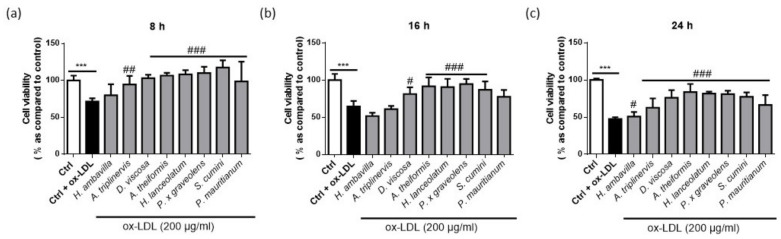
Cell viability of RAW 264.7 macrophages incubated with medicinal plant decoctions and cytotoxic concentration of ox-LDL for 8 (**a**), 16 (**b**) and 24 h (**c**). Data shown are means ± SD of three independent experiments. *** *p* < 0.005 as compared to control untreated cells (0 in white); ^#^
*p* < 0.05, ^##^
*p* < 0.01, ^###^
*p* < 0.005 as compared to cells treated with ox LDL (0 in black).

**Figure 3 nutrients-13-00280-f003:**
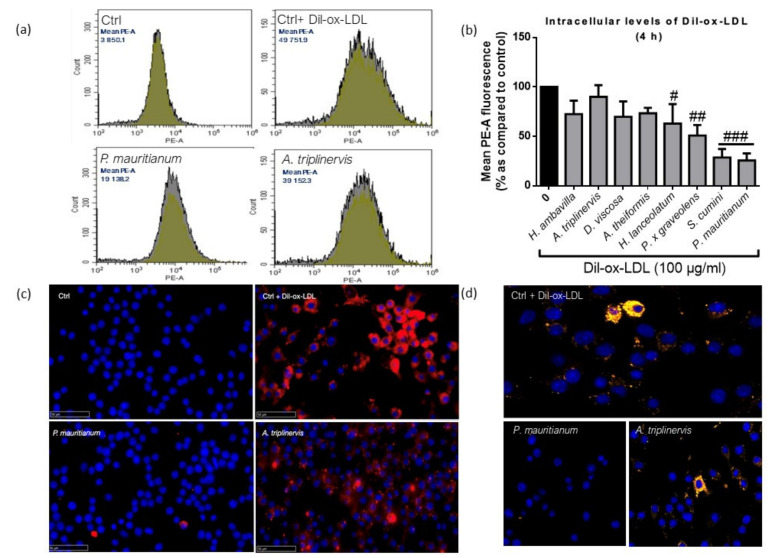
Dil-ox-LDL uptake by RAW 264.7 macrophages in the presence of medicinal plant decoctions analyzed by flow cytometry (**a**,**b**), nanozoomer scanner (**c**) and confocal microscopy (**d**), data shown in (**b**) are means ± SD of three independent experiments. ^#^
*p* < 0.05, ^##^
*p* < 0.01, ^###^
*p* < 0.005 as compared to cells treated with Dil-ox-LDL (0 in black). Blue color refers to DAPI (4′,6-diamidino-2-phenylindol) nuclear staining and red/yellow colors refer to Dil-ox-LDL, respectively, observed by fluorescence and confocal microscopy.

**Figure 4 nutrients-13-00280-f004:**
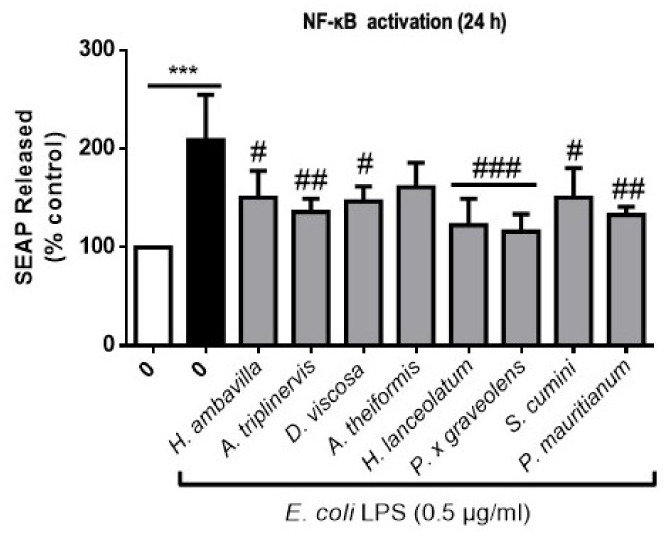
Modulation of NF-κB activation by medicinal plant decoctions (24 h). Data shown are means ± SD of three independent experiments. *** *p* < 0.005 as compared to cells untreated with lipopolysaccharides (LPS) (0 in white); ^#^
*p* < 0.05, ^##^
*p* < 0.01 and ^###^
*p* < 0.005 compared to cells treated with LPS (black bar).

## Data Availability

Not applicable.

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
