# Peer review of "Protective Effects of Medicinal Plant Decoctions on Macrophages in the Context of Atherosclerosis"

_nutrients, 2021, doi:10.3390/nu13010280_

Round 1
Reviewer 1 Report
The authors clearly report an in vitro study on the effects of medicinal plant decoctions on cytotoxicity, LDL accumulation and NFkB signaling of mouse derived macrophages. They demonstrate that the 8 different plant compounds that are traditionally used had some effects on the pathological processes in macrophages that occur during atherosclerosis disease. They conclude that the traditional herbal preparations could have some physiological basis and warrants further investigation.
Some specific points:
Abstract:
The final sentence should be toned down as the study did not investigate the effect of the herbal preparations on atherosclerosis itself, but on the one cell type that is involved in the atherosclerotic disease process.
Main text:
Based on their previous study on these medicinal products I believe that the authors could alter the wording of the final paragraph (lines 67-72) to include a hypothesis? Do they expect that all compounds will work similarly as their original paper showed varied amounts of polyphenols?
Provide details of French Blood agency (lines 86-87)
Add reference number for Ohakawa (line 102).
Provide details of Dil fluorescent probe
Add sentence to highlight that the macrophages are murine based (line 113). I was not immediately familiar with the purchased macrophages cell line, but assumed on first glance that they were human, given the human LDLs. Only on research into the cell lines did I discover that they were murine. Is there a scientific reasoning behind the use of mouse not humans? Does stimulation with ox-LDL from mouse have the same effects?
Section 2.7 (Neutral Red assay) (lines 120-129). There is no mention in this about the use of triton as positive controls for cytotoxicity that are shown in Figure 1. Could the authors add this information?
To ensure consistency of nomenclature change 100.103 to 100 x 103 (line 150)
Add version and details of Graphpad prism (line 160)
A range of concentrations of ox-LDL were tested (50-300 ug/mL) (line 172) However, there is no data from 50ug/mL, did this not have any effect? This could be an interesting point to discuss considering that 50 ug/mL has shown to affect vascular smooth muscle cell (VSMC) proliferation (Ding 2012, JRSoc Interface9(71):1233-40).
Add that 24h of 300ug/mL ox-LDL also decreases viability not just 200ug/mL (line 177)
Lines 185-186 would be better placed at line 193, after description of the time course of ox-LDL. In addition, all, not most, medicinal compounds showed protection at 24h.
There is no comment made about the time course (Figure 2), H ambavilla only shows protective effect at 24h, the other compounds tested appear to work earlier. This effect was also very small compared to the other compounds, would this couple of % more viable cells be physiologically relevant?
Greatest protective effects were observed for (lines 195-96). Is there any statistical evidence to support this theory? There is no reporting of statistical comparisons between the compounds in the manuscript.
Figure 2b shows that S cumini as well as P mauritianum seems to show similar reduced intracellular uptake of Dil-ox-LDL (line 214).
Did the authors do any type of quantification on the expression as shown in the representative images confocal images? This additional information could strengthen the manuscript.
Line 219-20 “This property could support traditionally described benefits of P mauritianum in cardiovascular disease”. This is a very bold statement that has not been tested in this in vitro study,
Lines 236-239. There is no data shown to support the statements made here that ox-LDL did not induce NFkB. Add data to Figure 3, or in the text. The authors show beneficial effect protecting NFkB activation of the medicinal components in response to the strong stimulus. What is the biological relevance to atherosclerosis if ox-LDL not effective. A brief literature search shows that this has previously been shown. What explanation do the authors have for their difference to published literature?
Conclusions:
The manuscript would benefit from a simple summary of the main findings of the study up front.
Singling out quercetin as the flavonoid (line 263) is not substantiated by the data presented in this manuscript. The authors previous manuscript outlines a wide range of flavonoids that could be involved.
Reference required for statement “ described to have cytoprotective effects” (line 265).
Figures:
Figure 1: There is inconsistency in the use of colours and shading in figure 1b and 1c. For example in Fig1b, control is grey, but in 1c it is represented by white bars.
Figure 2: a) Switching the medicinal flow cytometry plots would allow for an easier visualization that P mauritinum, is more similar to control than A triplinervis.
Data shown in (d) (line 227) is incorrect, the data is shown in panel b.
Include what red / yellow and blue represent in the images to aid the reader to interpret.
Author Response
Please see the attachement

Reviewer 2 Report
Dear Editor,
I carefully read the manuscript by Checkouri et al., which is overall interesting, though needs to be carefully revised.
My comments and suggestions for the authors are the following:
- Some authors' claims are not supported by a source. In particular, I suggest authors to include a references before the stop at line 44 and a references before the stop at line 45. Authors should consider to refer also to doi: 10.1038/s41598-018-20425-x and doi: 10.1016/j.ijcard.2018.03.077.
- English language needs to be carefully revised throughout the manuscript. In particular, authors are suggested to to pay particular attention to verbal concordances (e.g. I suggest the authors to use the past simple also in the "Statistics" paragraph).
- As regard "Statistics", I suggest the authors to present normally distributed parameters as mean plus/minus standard deviation and non-normally distributed parameters as median and confidence interval (or variation range).
- Study's limitations should be discussed by the authors.
- Line 265: A reference is missing before the stop.
- In general, I suggest authors to update references 3, 4 and 5, that are quite obsolete.
Author Response
Please see the attachement.

Round 2
Reviewer 2 Report
Dear Editor,
I carefully read the revised version of the manuscript, which is significantly improved in comparison with the original one.